# The serogroup B meningococcal outer membrane vesicle-based vaccine 4CMenB induces cross-species protection against *Neisseria gonorrhoeae*

Isabelle Leduc[1☯], Kristie L. Connolly[1☯], Afrin Begum[1], Knashka Underwood[1], Stephen Darnell[1], William M. Shafer[2,3], Jacqueline T. Balthazar[2], Andrew N. Macintyre[4], Gregory D. Sempowski[4], Joseph A. Duncan[5,6], Marguerite B. Little[5], Nazia Rahman[7], Eric C. Garges[8], Ann E. Jerse[1]*

1 Department of Microbiology and Immunology, Uniformed Services University, Bethesda, Maryland, United States of America, 2 Department of Microbiology and Immunology and The Emory Antibiotic Resistance Center, Emory University School of Medicine, Atlanta, Georgia, United States of America, 3 Laboratories of Bacterial Pathogenesis, Atlanta Veterans Affairs Medical Center, Decatur, Georgia, United States of America, 4 Duke Human Vaccine Institute, Duke University School of Medicine, Durham, North Carolina, United States of America, 5 Department of Pharmacology, University of North Carolina, Chapel Hill, North Carolina, United States of America, 6 Division of Infectious Diseases, Department of Medicine, University of North Carolina, Chapel Hill, North Carolina, United States of America, 7 Infectious Disease Clinical Research Program, Department of Preventive Medicine and Biostatistics, Uniformed Services University, Bethesda, MD, United States of America, 8 Department of Preventive Medicine and Biostatistics, Uniformed Services University, Bethesda, Maryland, United States of America

☯ These authors contributed equally to this work.
* ann.jerse1@usuhs.edu

**Data Availability Statement:** All raw data files are available from the Open Science Framework database (DOI 10.17605/OSF.IO/QU4VM).

## Abstract

There is a pressing need for a gonorrhea vaccine due to the high disease burden associated with gonococcal infections globally and the rapid evolution of antibiotic resistance in *Neisseria gonorrhoeae* (*Ng*). Current gonorrhea vaccine research is in the stages of antigen discovery and the identification of protective immune responses, and no vaccine has been tested in clinical trials in over 30 years. Recently, however, it was reported in a retrospective case-control study that vaccination of humans with a serogroup B *Neisseria meningitidis* (*Nm*) outer membrane vesicle (OMV) vaccine (MeNZB) was associated with reduced rates of gonorrhea. Here we directly tested the hypothesis that *Nm* OMVs induce cross-protection against gonorrhea in a well-characterized female mouse model of *Ng* genital tract infection. We found that immunization with the licensed *Nm* OMV-based vaccine 4CMenB (Bexsero) significantly accelerated clearance and reduced the *Ng* bacterial burden compared to administration of alum or PBS. Serum IgG and vaginal IgA and IgG that cross-reacted with *Ng* OMVs were induced by 4CMenB vaccination by either the subcutaneous or intraperitoneal routes. Antibodies from vaccinated mice recognized several *Ng* surface proteins, including PilQ, BamA, MtrE, NHBA (known to be recognized by humans), PorB, and Opa. Immune sera from both mice and humans recognized *Ng* PilQ and several proteins of similar apparent molecular weight, but MtrE was only recognized by mouse serum. Pooled sera from 4CMenB-immunized mice showed a 4-fold increase in serum bactericidal$_{50}$ titers

**Funding:** This work was supported by an interagency agreement between the National Institutes of Health (NIH), National Institute of Allergy and Infectious Diseases (NIAID), and Uniformed Services University (AAI14024, A.E.J) and a Defense Health Agency immunizations Healthcare Division grant (IHBISP_18_017, E.C.G). The Immunology Unit of the Duke Regional Biocontainment Laboratory received partial support for construction from NIH/NIAID (UC6-AI058607) G.S. W.M.S. was supported by a Senior Research Career Award from the Biomedical Laboratory Research and Development Service of the Department of Veterans Affairs. M.B.L. was supported by an NIH Basic Immune Mechanisms Training Grant (T32AI007273-33) and J.T.B. was supported by NIH/NIAID U19-AI144180 (A.E.J). NIAID url: www.niaid.nih.gov NIAID played a role in the study design as part of the interagency agreement. The funders had no role in study design, data collection and analysis, decision to publish, or preparation of the manuscript.

**Competing interests:** The authors have declared that no competing financial interests exist.

against the challenge strain; in contrast, no significant difference in bactericidal activity was detected when sera from 4CMenB-immunized and unimmunized subjects were compared. Our findings directly support epidemiological evidence that *Nm* OMVs confer cross-species protection against gonorrhea, and implicate several *Ng* surface antigens as potentially protective targets. Additionally, this study further defines the usefulness of murine infection model as a relevant experimental system for gonorrhea vaccine development.

## Author summary

Eighty-seven million *Neisseria gonorrhoeae (Ng)* infections occur globally each year and control of gonorrhea through vaccination is challenged by a lack of strong evidence that immunity to gonorrhea is possible. This contention was recently challenged by epidemiological evidence suggesting that an outer membrane vesicle (OMV) vaccine from the related species *Neisseria meningitidis* (*Nm*) protected humans against gonorrhea. Here we provide experimental evidence in support of this hypothesis by demonstrating that a licensed, modified version of this *Nm* OMV-based vaccine accelerates clearance of *Ng* in a mouse infection model. These results confirm the possibility cross-species protection and are important in that they support the biological feasibility of vaccine-induced immunity against gonorrhea. We also showed that several *Ng* outer membrane proteins that may be protective targets of the vaccine are recognized by antiserum from vaccinated mice, at least two of which (PilQ and NHBA) are also detected with sera from 4CMenB-immunized human subjects. Our demonstration that a vaccine that may reduce the risk of gonorrhea in humans protects mice against *Ng*, a highly host-restricted pathogen, also validates the mouse model as a useful tool for guiding the development of other candidate gonorrhea vaccines.

## Introduction

An estimated 87 million new gonorrheal infections occur each year worldwide [1] and rates are rising globally, with a 67% increase in reported infections in the U.S. between 2014 and 2018 [2]. Caused by the Gram-negative bacterium *Neisseria gonorrhoeae (Ng)*, gonorrhea is associated with significant morbidity and mortality that disproportionately affects women and newborns. Lower urogenital tract infections can ascend to cause endometritis and salpingitis in females, and less frequently, epididymitis and orchitis in males. *Ng* pelvic inflammatory disease can be asymptomatic or acute, and is associated with ectopic pregnancy, infertility and chronic pelvic pain. Disseminated gonococcal infection can occur in either gender [3]. Transmission of gonorrhea to neonates from infected mothers can cause acute neonatal conjunctivitis [4] and there is a clear association between maternal gonorrhea, low-birth weight and pre-mature delivery [5]. The impact of gonorrhea on human health is amplified by its role in increasing both transmission and susceptibility to the human immunodeficiency virus (HIV) [6,7].

Gonorrhea is classified as an urgent public health threat due to decreasing susceptibility to the last remaining reliable monotherapy for gonorrhea, the extended-spectrum cephalosporins. Dual therapy with high-dose ceftriaxone and azithromycin is currently recommended for empirical treatment of gonorrhea in many countries. However, *Ng* susceptibility to these antibiotics continues to decrease world-wide [8], and alarmingly, treatment failures due to strains that are resistant to these antibiotics have been reported [9,10]. New antibiotics are under development [11,12]; however, the evolutionary success of the gonococcus in outrunning public health efforts to contain it through treatment has reinvigorated the call for a gonorrhea vaccine [13,14].

Early vaccine research was challenged by the discovery that several *Ng* surface molecules are phase or antigenically variable. There was also no animal model other than chimpanzees for systematic testing of immunogens in challenge studies, and two published clinical trials using a killed whole cell vaccine [15] or purified pili [16] were unsuccessful despite earlier small in-house studies that showed protection from urethral challenge in human male volunteers [17]. Since this time, several conserved and semi-conserved vaccine antigens that elicit bactericidal antibodies or inhibit target function have been identified, some of which show protection in a well-characterized mouse genital tract infection model [13]. How well the mouse model predicts vaccine efficacy in humans is not known, however, due to the strict host-specificity of *Ng* and a lack of information on correlates of immune protection in humans. There is little immunity to natural infection in humans and mice [18], and there is growing evidence that the adaptive response to *Ng* infection is suppressed. As recently reviewed by Lovett and Duncan [19], human humoral immune responses to *Ng* to infection are modest at best and the analysis thereof is complicated by pre-existing antibodies to carbohydrate and protein surface antigens that are induced by commensal *Neisseria sp*., although antibodies to some antigens are increased by infection. Human cellular responses to *Ng* infection are less well studied, but appear to be driven by a Th17 pro-inflammatory response. Th1 responses, in contrast, appear suppressed [20], and several pathways that result in reduced antigen presentation and, or inhibition of T cell responses to *Ng* have been identified using human immune cells and experimentally infected mice [21–25].

Recent epidemiological evidence, however, suggests immunity to gonorrhea can be achieved in humans through vaccination with outer membrane vesicles (OMVs) of the related species, *Neisseria meningitidis* (*Nm*). In this cross-sectional study, vaccination of individuals with the serogroup B meningococcal vaccine MeNZB, which consisted of OMVs from an endemic New Zealand strain, was associated with a reduced rate of gonorrhea in adolescents and adults aged 15–30 years old [26]. Using cases of chlamydia as a control, the estimated effectiveness of this meningococcal vaccine against gonorrhea was predicted to be 31%. These data are the first controlled evidence in humans in over 40 years that vaccine-induced protection against gonorrhea is possible. A similar finding was suggested by epidemiological studies on *Nm* OMV vaccines in Cuba and Norway [27].

To directly test the hypothesis that *Nm* OMV-based vaccines induce cross-species protection against *Ng*, here we evaluated the *in vivo* efficacy of the licensed 4CMenB ("4 Component Meningitis B"; Bexsero) vaccine in a female mouse model of *Ng* lower genital tract infection. 4CMenB consists of *Nm* OMVs from the *Nm* strain used in the MenNZB vaccine and five recombinant *Nm* proteins [28], only one of which, the neisserial heparin-binding antigen (NHBA), is a feasible vaccine target for gonorrhea [29,30]. Our results show that 4CMenB significantly reduces the *Ng* bioburden, accelerates clearance of infection, and induces antibodies that recognize *Ng* outer membrane proteins, several of which are promising vaccine targets. These findings are consistent with epidemiological data that suggest cross-species protection against gonorrhea is possible and validate the gonorrhea mouse model as a useful experimental system for developing vaccines against this human disease.

## Results

### Optimization of the immunization regimen to induce serum and vaginal antibodies

The recommended dosing regimen for 4CMenB in humans is two 500 µL doses given intramuscularly, four weeks apart. As a preliminary step for mouse immunization/challenge studies, we immunized BALB/c mice with 20, 125, or 250 µL of the 4CMenB vaccine on days 1 and

28 by the subcutaneous (SC) or intraperitoneal (IP) routes to assess safety and immunogenicity. A dose-response in serum IgG1 titers against the 4CMenB vaccine components was detected by ELISA in IP- and SC-immunized mice (S1A Fig), and IgG2a titers were higher in mice given 100 μL or 250 μL compared to 20 μL (S1B Fig). This dose response was mirrored in Western blots using anti-IgG secondary antibody against whole-cell lysates of *Nm* and six different *Ng* strains (S1C Fig). Sera from control mice given PBS or Alum adjuvant alone did not recognize any *Nm* or *Ng* proteins. No adverse effects were observed following IP injection. Nodules formed at the injection site in SC-immunized mice for all doses given, which resolved over time.

The MenNZB human epidemiology study reported by Petousis-Harris, et *al.* [26] was based on subjects who received a 3-dose regimen separated by one month. Upon demonstrating that the 250 μL dose of 4CMenB was well-tolerated and induced the highest serum antibody titers, we added a third immunization in subsequent mouse immunization/challenge experiments. Mice were given 250 μL of the formulated vaccine three times by IP or SC injection; controls received Alum or PBS (IP). A 3-week interval between immunizations was used to avoid increasing the age of the mice before challenge, which can reduce susceptibly to *Ng*. Significantly higher titers of *Ng*-specific serum total Ig, IgG1, and IgG2a ($p < 0.004$) but not IgA were detected in SC- and IP-immunized mice against *Ng* OMVs compared to control groups that received Alum or PBS on day 52 (Fig 1A, 1B, 1C and 1D). Total Ig, IgG1 and IgG2a titers were further elevated by the third immunization in both the IP- and SC-immunized groups (day 31 versus day 52) ($p \leq 0.05$). The IgG1/IgG2a ratio was significantly lower for IP-immunized mice on day 31, but similar to SC-immunized mice on day 52 (Fig 1E) due to a marked increase in IgG2a titers in the SC-immunized group after the third immunization (Fig 1C). Vaginal total Ig and IgG1 ($p < 0.0001$), but not IgG2a or IgA were significantly elevated in vaginal washes collected after the second immunization compared to control groups in IP-immunized mice, but not SC-immunized mice (Fig 1F, 1G, 1H and 1I). Vaginal washes were not collected after the third immunization to avoid altering the vaginal microenvironment before bacterial challenge and cessation of the LeBoot effect, which would increase the number of mice in the undesired stages of the estrous cycle at the time of challenge [31]. We conclude that a half human-dose of 4CMenB is well-tolerated in mice and that a dosing regimen similar to that used in the New Zealand study elicits systemic and mucosal humoral immune responses that are cross-reactive against *Ng*.

## 4CMenB-immunized mice clear *Ng* infection significantly faster and have a reduced bioburden following vaginal challenge

To assess the protective efficacy of 4CMenB against *Ng*, we challenged 4CMenB-immunized and control mice with *Ng* strain F62 three weeks after the third immunization and quantitatively cultured vaginal swabs for *Ng* over seven days. In combined data from two independent experiments, IP-immunized mice exhibited a significantly faster clearance rate (p ≤0.0001) (Fig 2A) and lower bioburden compared to control groups given PBS or Alum alone (p <0.05 and ≤0.01, respectively) (Fig 2B and 2C). Data from each individual experiment also showed significantly faster clearance for both immunized groups compared to control mice, (S2A and S2D Fig). The bioburden in the IP-immunized group was significantly lower than that of the Alum only group in both experiments, while the difference in the bioburden in SC-immunized mice compared to Alum was significant only in the repeat experiment (p < 0.0001) (S2B and S2E Fig). Unimmunized mice from the first experiment had a significantly higher AUC compared to 4CMenB-immunized mice by either route (S2C Fig), however, in the second study only a significant reduction between the SC-immunized and alum groups was observed

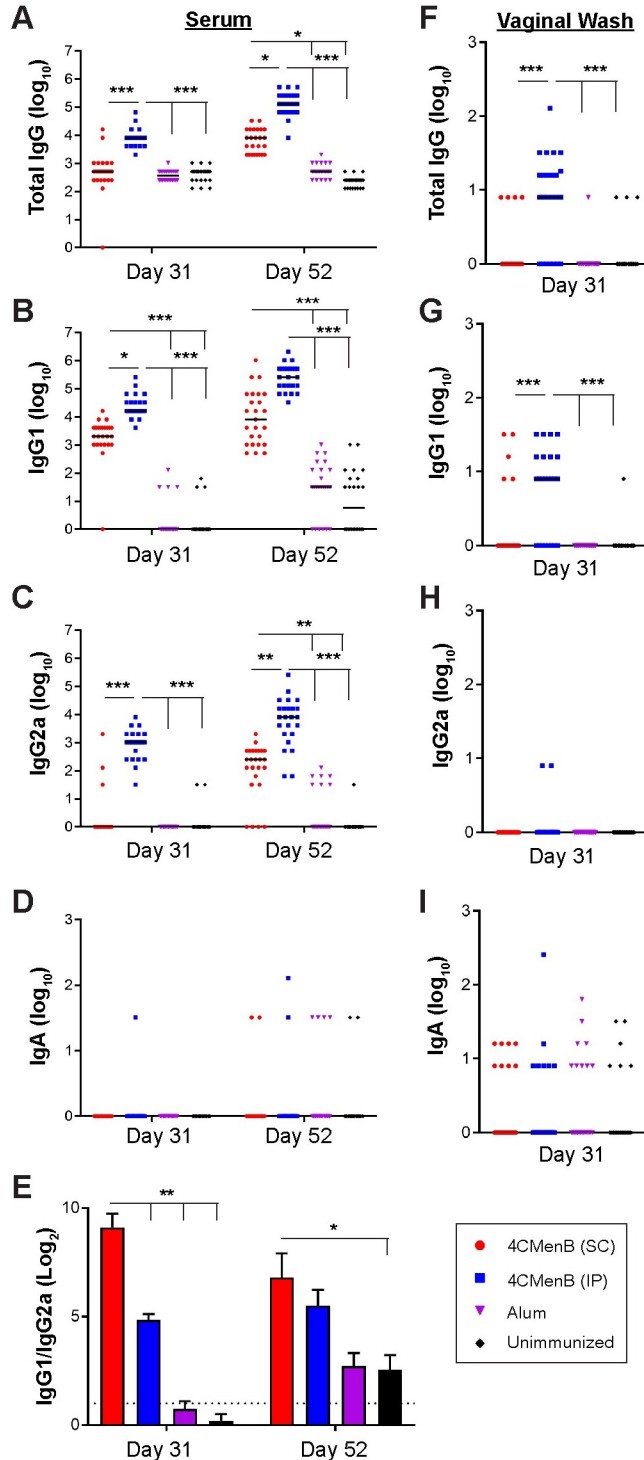

**Fig 1. Serum and vaginal antibody titers against F62 OMVs from 4CMenB-immunized mice.** Groups of 25 BALB/c mice were immunized three times, three weeks apart with 250 µl or 4CMenB by the IP or SC routes or with PBS or Alum (IP route). Serum and vaginal antibody titers on day 31 and day 52 (ten days after the 2nd and 3rd immunization, respectively) against F62 OMVs were measured by ELISA. Shown are serum **(A)** total Ig, **(B)** IgG1, **(C)** IgG2a, **(D)** IgA and **(E)** IgG1/IgG2a ratios for on days 31 and 52 (left column). Vaginal **(F)** total Ig, **(G)** IgG1, **(H)** IgG2a and **(I)** IgA on day 31 are shown in the right column. No difference was found in any sample or Ig tested between control animals receiving PBS and those receiving alum only. \*, p<0.05; \*\*, p<0.01; \*\*\*, p<0.0001. Results from the repeat experiment were similar.

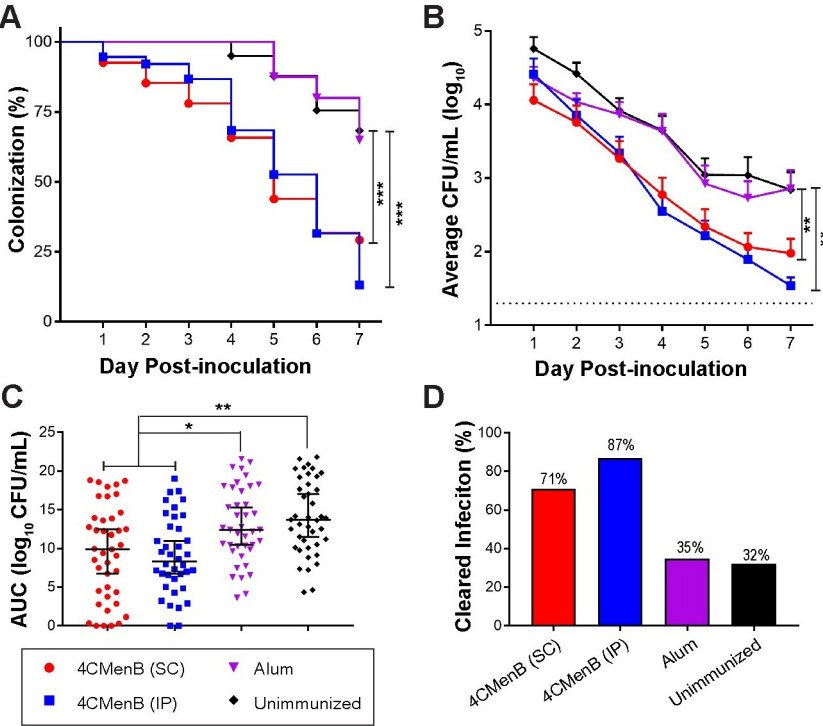

**Fig 2. 4CMenB has *in vivo* efficacy against *Ng*.** Mice were immunized three weeks apart with 250-μL doses of 4CMenB by the IP (blue) or SC (red) route or with PBS (black) or alum (purple) by the IP route and challenged with *Ng* strain F62 three weeks after the final immunization. Shown are the combined data from two independent trials (total n = 38–41 mice/group). **(A)** Percentage of culture-positive mice over time; **(B)** Average CFU per ml of a single vaginal swab suspension; **(C)** total bioburden over 7 days expressed as area under the curve; **(D)** Percentage of mice that cleared infection by day 7 post-challenge. *, p < 05; **, p < 0.01; ***, p < 0.0001.

(S2F Fig). Combined data from the two experiments showed that 70% and 88% of mice given 4CMenB by the SC and IP routes, respectively, cleared infection by day 7 compared to 25–30% of mice given alum or PBS (Fig 2D).

A peak vaginal PMN influx beginning on day 4 post-bacterial challenge was observed in all groups, and there was no difference in percentage of PMNs among experimental groups over time (S3 Fig). We also evaluated complement-dependent bactericidal activity of pooled serum from each group against *Ng* strain F62, the serum-sensitive challenge strain, and against the serum-resistant strain FA1090, using normal human serum as the complement source. The bactericidal$_{50}$ titers were 1:480 and 1:240, respectively, which were 4-fold greater than that of pooled serum from the unimmunized group (Fig 3). We conclude that 4CMenB reproducibly accelerates clearance of *Ng* from the murine genital tract and lowers the bioburden over time and that opsonophagocytosis and complement-mediated bacteriolysis may contribute to the protection.

## 4CMenB-Induced serum and vaginal antibodies cross-react with several *Ng* OMV proteins

To examine the cross-reactivity of 4CMenB-induced antibodies against *Ng* surface proteins, we performed western immunoblots against OMVs from the challenge strain and five other *Ng* strains that are geographically and temporally distinct in their isolation. Pooled antiserum from mice immunized twice with the 250 μL dose by either the IP or SC routes (250IP or

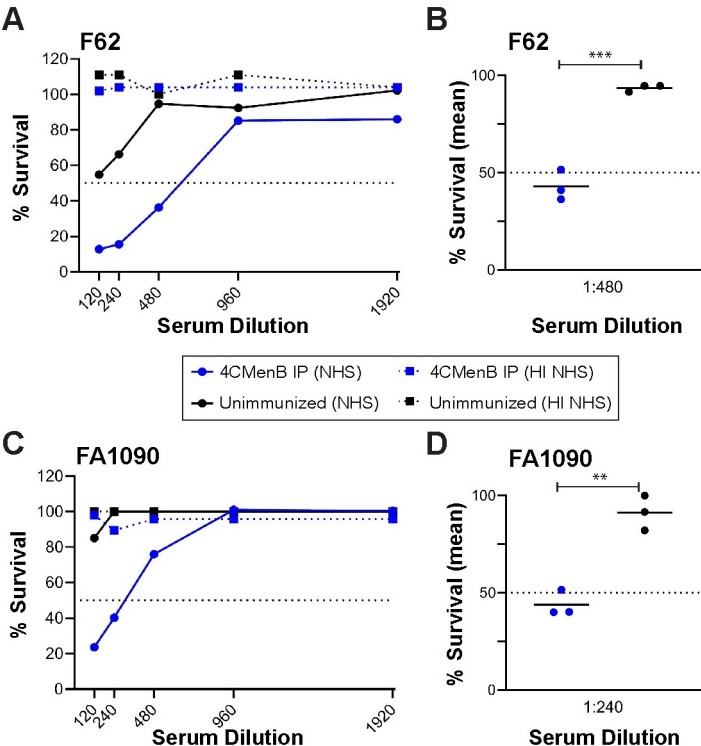

**Fig 3. Pooled antiserum from 4CMenB-immunized mice is bactericidal against a serum-sensitive and a serum-resistant *Ng* strain.** Serial dilutions of pooled serum from mice vaccinated with 250 μl of 4CMenB (blue lines) or Alum alone (black lines) by the IP route were incubated with $10^4$ CFU of the challenge strain F62 or strain FA1090 in microtiter plates as described in the Methods. After 5 min, NHS or heat-inactivated (HI)-NHS (final concentration, 10%) was added. After 55 min incubation at 37˚C, the number of viable *Ng* in each well was determined by duplicate culture on GC agar. **(A, C)** Recovery of F62 **(A)** or FA1090 **(C)** from each condition expressed as the number of CFU from wells incubated with test serum divided by the number recovered from wells containing PBS instead of test serum, X times 100. Solid lines indicate NHS was used as the complement source; dotted lines represent data from wells tested in parallel with HI-NHS. The dotted line at 50% survival was drawn to identify the bactericidal$_{50}$ titers. **(B, D)** Recovery of viable F62 **(A)** or FA1090 **(B)** from a 1:480 or 1:240 dilution of serum, respectively, from 4CMenB-immunized mice or unimmunized mice for each of three independent experiments, expressed as above. A significant difference was observed between the immunized and unimmunized serum (**, p < 0.01; ***, p < 0.0001).

250SC, respectively) recognized four prominent bands in fractionated OMV preparations all six strains: a high molecular weight (HMW) band > 220 kD, a doublet with bands of apparent molecular weight of 97 and 94 kDa, and a 55 kDa band (Fig 4). Several low intensity bands between 26 and 36 kDa were also recognized in several of the strains. Reactivity of the 250SC antiserum was weaker than the 250IP antiserum, which likely reflects the lower titers of this antiserum. Consistent with ELISA data, serum reactivity as assessed by band intensity was increased by a third immunization (Fig 4A), and additional bands were recognized including several bands in the 30–35 kDa range. A similar recognition pattern was observed on blots incubated with pooled vaginal washes and sera collected 10 days after the third immunization from immunized but unchallenged mice followed by anti-mouse IgG or anti-mouse IgA (Fig 4B; compare lanes 1 and 2 with lane 5 in each blot). These results also show that while vaginal titers were low after the second immunization as measured by ELISA (Fig 1), vaccine-induced vaginal antibodies were readily detectable by immunoblot after a third immunization.

We also examined the reactivity of the immune serum against *Ng* lipooligosaccharide (LOS). Neisserial LOS is a branched structure that consists of oligosaccharide extensions from

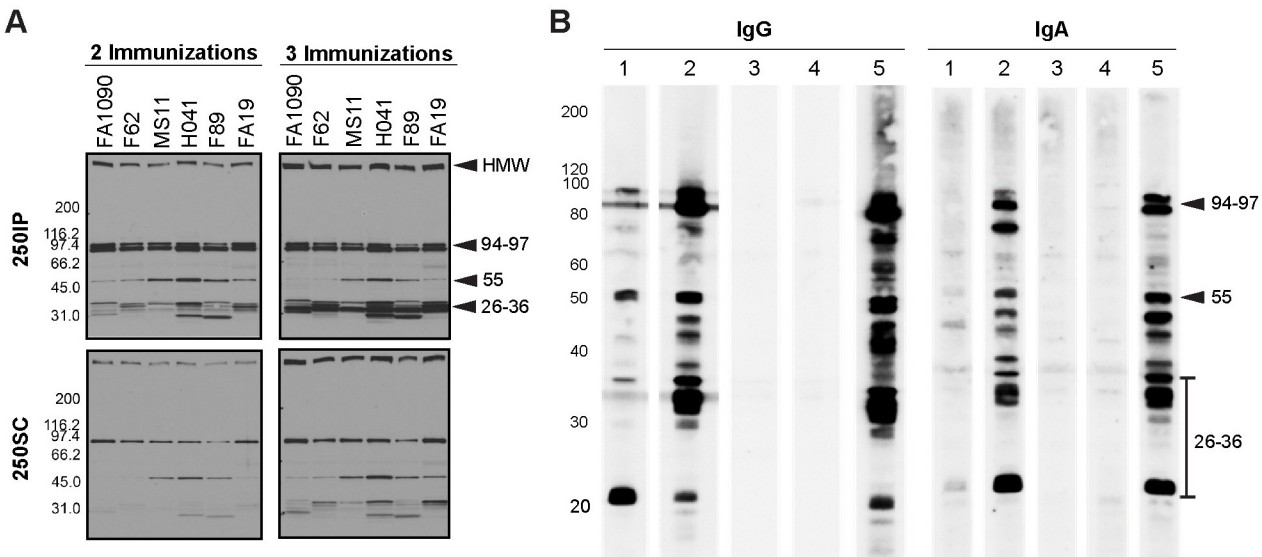

**Fig 4. Serum and vaginal antibodies from 4CMenB-immunized mice recognize *Ng* outer membrane proteins by western immunoblot. (A)** Pooled antiserum from mice immunized with 250 μL of 4CMenB by the IP (250IP, upper panels) or SC (250SC, lower panels) route were tested against OMVs (app. 20 μg per lane) from 7 different *Ng* strains fractionated on 4–20% Tris-glycine gels by western blot (1:10,000 dilution of primary antisera) followed by secondary anti-mouse IgG-HRP. A boosting effect is observed when comparing the band intensities for serum collected after 2 and 3 immunizations. **(B)** Pooled vaginal washes from immunized or control mice collected after the third immunization tested against OMVs from the F62 challenge strain by western blot (1:100 dilution), followed by secondary anti-mouse IgG-HRP or anti-mouse IgA-HRP. Pooled vaginal washes were from mice given: (1) SC250; (2) IP250; (3) PBS; (4) Alum only. (5) 250IP mouse serum (1: 10,000) used for comparison. The band recognition pattern was similar for blots incubated with serum (lanes 5) or vaginal washes (lanes 1–4), and vaginal washes from IP-immunized mice were more strongly reactive than from SC-immunized mice. All lanes were equally loaded, as determined by Ponceau S staining. Shown are representative results from at least 2 separate experiments with identical results.

the core oligosaccharide called the α- and β-chains. An additional extension called the γ-chain is present in some strains [32]. Different LOS species can be produced within a strain due to phase variable expression of the glycosyltransferase genes *lgtA*, *lgtC*, *lgtD* and *lgtG*, which can reveal or mask LOS epitopes [33,34]. To test the cross-reactivity of 4CMenB antisera against *Ng* LOS, we fractionated crude LOS extracts by gel electrophoresis from four laboratory and thirteen clinical *Ng* isolates isolated between 1991 and 2019. and examined the reactivity with monoclonal antibodies (3F11, 4C4 and 2C7) that recognize known epitopes within *Ng* LOS [35] or with the 4CMenB antiserum by Western blot. All of the strains produced an LOS that bound one or more of monoclonal antibodies (S4A Fig). In contrast, LOS was not recognized by the 4CMenB antiserum in thirteen of the seventeen strains tested (S4B Fig). The four strains with cross-reactive LOS (MS11 and three clinical isolates LGB24, NMCSD322 and NMCSD6364) produced one or two LOS species that were recognized by the 4CMenB antiserum, but not by serum from mice given Alum adjuvant alone (S4B and S4C Fig). Schneider *et al*. [36] demonstrated that long-chain LOS species are selected during urethral infection in men and Rice and colleagues have shown that the phase variable 2C7 LOS epitope is expressed among a majority of clinical isolates [35]. Therefore, to test investigate whether the anti-4CMenB-reactive LOS epitope is perhaps selected *in vivo*, we infected mice with H041, which did not produce a cross-reactive LOS. No 4CMenB-reactive LOS species were detected in LOS preps from pooled vaginal H041 isolates cultured on days 2 and 5 post-inoculation (S4D Fig). We conclude that while cross-reactive antibodies to *Ng* LOS epitopes are induced by 4CMenB, the epitopes do not appear to be shared by a majority of *Ng* strains.

## 4CMenB induces antibodies against promising *Ng* vaccine targets

To identify the proteins recognized by 4CMenB-induced antisera, we fractionated OMVs from *Ng* strain F62 on two separate gels. One was stained with a G-250 Coomassie stain for mass spectrometry analysis, while the other was used for Western blotting with the 250IP antiserum. The blot and gel were aligned and the reactive bands identified by molecular weight and band intensity. Bands indicated by the numbered arrows (Fig 5A), which correspond to the most intensely recognized bands in the Western blot (Fig 5B), were submitted for mass spectrometry analysis (Table 1). The HMW band at the top of the gel was identified as PilQ, which is a protein that forms a dodecamer through which gonococcal pili extend [37]. Mass spectrometry analysis identified two proteins in band 2 (97 kDa): an elongation factor and a phosphoenolpyruvate. Band 3 (94 kDa) also contained two proteins: BamA, an Omp85 homologue involved in the biogenesis of OM proteins (OMPs) [38], and a methyltransferase. Band 4 (55 kDa) was identified as MtrE, the OM channel of the three different gonococcal active efflux pump systems [39]. The 36 kDa protein (band 5) was identified as PorB, and the 32 kDa band (band 6), as Opa. In summary, we identified eight proteins from six cross-reactive bands, five of which are known surface-exposed *Ng* antigens.

To examine whether 4CMenB-induced antibodies bind native PilQ and MtrE on the gonococcal surface, we next performed immunoprecipitations using 4CMenB-induced mouse antisera and live FA1090 and MS11 bacteria. Antigen-antibody complexes were solubilized in detergent, retrieved using protein A/G agarose, and subjected to non-denaturing Western blotting with the 250IP antiserum. Whole cell lysates and OMVs were run in parallel and exhibited the same reactive band pattern shown in Figs 4A and 5B. Serum from the alum-alone group pulled down non-specific proteins smaller than 30 kDa that did not align with bands in lanes containing whole cell lysates or OMVs. In contrast, the 250IP antiserum pulled down two proteins from both strains: a HMW protein, possibly PilQ, and a ~ 55 kDa protein (Fig 6). We hypothesized the 55 kD protein was MtrE based on the mass spectrometry data (Table 1) and the greater intensity of this band in western blots against OMVs from strains MS11, H041 and F89, which carry one or more *mtr* mutations that cause increased production of the MtrCDE efflux pump (Fig 4A).

To confirm the identity of the two proteins, we included single isogenic *pilQ* and *mtrE* mutants, and the results confirmed the identity of the two immunoprecipitated proteins as PilQ and MtrE. The HMW band but not the 55-kDa protein was absent from the sample in which mutant strain FA1090*pilQ* was incubated with the antisera. Inversely, immunoprecipitation with mutant MS11*mtrE-* did not yield a band around 55 kDa, but did retain binding to the HMW protein PilQ (Fig 6). We conclude that 4CMenB induces antibodies recognize the native conformation of *Ng* PilQ and MtrE on the gonococcal surface.

## Antibodies that cross-react with PilQ and NHBA are induced in both vaccinated humans and mice

To compare the reactivity of serum from 4CMenB-immunized mice with that of humans, we performed western blots with pooled immunized mouse serum and pooled sera from study participants who had received 2 doses of the 4CMenB vaccine (Fig 7). Several *Ng* OMV proteins were recognized by the 4CMenB-specific human antisera (Fig 7B) compared to unimmunized serum (Fig 7C), and comparison of the blots showed recognition of at least seven bands of similar apparent molecular weight by both human and murine sera, including a prominent band at ~21 kD, bands in the 26–31 kD region, a faint band at ~37 kD, a ~46 kD band, bands in the 66–77 kD region and a ~ 83 kD band (Fig 7A and 7B). Notably, MtrE was recognized by only by antiserum from 4CMenB-immunized mice. The recognition pattern produced by

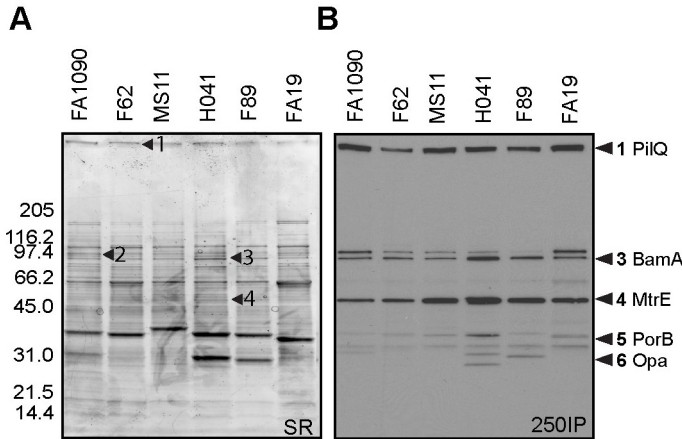

**Fig 5. PilQ, MtrE, porin and Opa are recognized by 4CMenB antisera.** OMVs (app. 20 μg) from 6 *Ng* strains (Table 2) were subjected to SDS-PAGE on a 4–20% Tris-glycine gel and **(A)** stained with sypro ruby or **(B)** transferred to PVDF for western blot with the 250IP antiserum. The stained gel was aligned with the Western blot, and corresponding bands were digested and analyzed by mass spectrometry. The numbers indicated with arrows on each panel correspond to the same numbers on the Western blots except for bands 5 and 6, which were excised from a different gel but are indicated on the western based on the banding patterns. Proteins identified are described in Table 1. Among the proteins identified, known surface-exposed outer membrane proteins are: (1) PilQ, (3) BamA, (4) MtrE, (5) PorB, and (6) Opa.

individual sera from seven immunized persons was similar to that of the pooled serum, with some differences in band intensity present among the samples (Fig 8A). Most individual unimmunized serum samples recognized a ~ 62 kD and 119 kD band that were also recognized by sera from immunized subjects (but not by the immune mouse sera), and is most likely due to pre-existing antibody and not antibodies induced by vaccination (Fig 8B). A second set of immunized and unimmunized serum samples were tested against blots of gels that were

**Table 1. Identification of protein bands recognized by 4CMenB antisera as determined by mass spectrometry.**

| Sample (band) | Protein name | Database Accession ID[a] | MW (Da) | PeptideCount[b] | MS & MS/MS Score[c] | Peptide sequenced ion score[d] | Scoring Threshold[e] |
|---|---|---|---|---|---|---|---|
| 1 | Type IV pilus biogenesis and competence protein PilQ | Q5FAD2 | 77903 | 9 | 556 | 482 | 55 |
| 2 | Elongation factor G | B4RQX2 | 77124 | 12 | 852 | 722 | 55 |
| 2 | Phosphoenolpyruvate synthase | KLS49216 | 87167 | 10 | 666 | 557 | 55 |
| 3 | Outer membrane protein assembly factor BamA | Q5F5W8 | 87888 | 16 | 617 | 481 | 55 |
| 3 | 5-methyltetrahydropteroyltriglutamate—homocysteine methyltransferase | Q5F863 | 85030 | 6 | 227 | 204 | 55 |
| 4 | Multidrug transporter MtrE | Q5F726 | 50382 | 10 | 452 | 317 | 55 |
| 5 | Porin (PorB) | YP_208842 | 35516 | 16 | 969 | 814 | 55 |
| 6 | PII/Opa | Q51014 | 31429 | 10 | 670 | 568 | 55 |

[a]For the protein sequence, search this ID at "http://www.ncbi.nlm.nih.gov", under the parameter to "Protein";

[b]Number of observed peptides matching the theoretical digest of the identified protein;

[c]Combined score of the quality of the peptide-mass fingerprint match and MS/MS peptide fragment ion matches (if MS/MS data was generated);

[d]Score of the quality of MS/MS peptide fragment ion matches only (if MS/MS data was generated);

[e]Significant score threshold. A hit with an "MS & MS/MS score" or an "Ion score" above this value is considered a significant identification. p<0.05 for the given species database. Only database search hits with "MS and MSMS scores" above this value are reported.

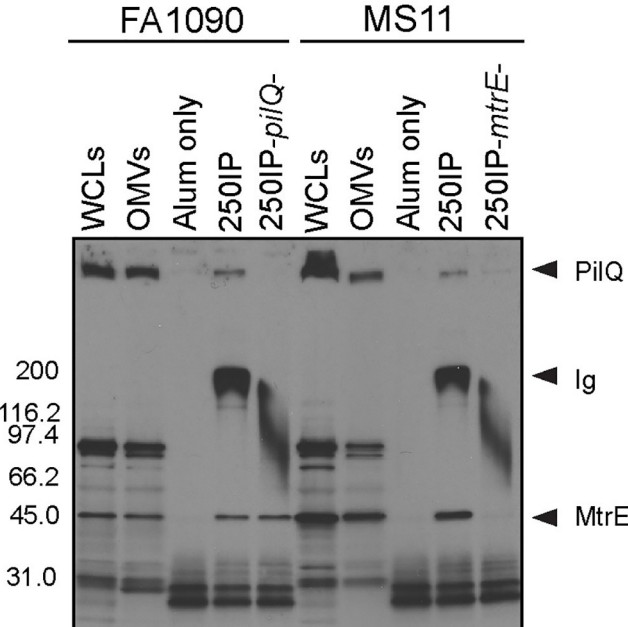

**Fig 6. 4CMenB -induced antibodies bind PilQ and MtrE at the surface of viable *Ng* FA1090 and MS11 bacteria.**
Immunoprecipitations were performed with wild-type strains FA1090 and MS11 and their isogenic *pilQ* and *mtrE*
mutants using antisera from mice immunized with 250 μL of 4CMenB via the IP route (250IP) or given alum only
(negative control). Bacterial components bound by 4CMenB-induced antisera were subjected to SDS-PAGE (non-
denaturing conditions, 4–20% Tris-glycine) and Western blotting with 4CMenB 250IP antiserum. WCLs, total cellular
proteins; OMVs, outer membrane vesicles; *pilQ*-, FA1090Δ*pilQ*; *mtrE*-, MS11Δ*mtrE*. Data shown are representative of
at least 2 separate experiments with identical results. The wide band around 200 kDa corresponds to the antibodies
within the test antisera that are present in the antigen-antibody complexes and pulled down with the protein A/G
agarose.

fractionated for a longer period to look for recognition of PilQ, which due to its size does not
always enter the gel or transfer efficiently. An intense band > 220 kD was recognized by sera
from four of five immunized individuals (Fig 8C) that was not seen in sera from two unimmu-
nized individuals (Fig 8D). The molecular weight of this band is consistent with PilQ, which
was also detected by serum from immunized mice (Figs 4, 5 and 6) and confirmed by mass
spectrophotometry to be PilQ (Table 1).

To identify additional proteins that appeared to be recognized by both mice and humans,
we investigated whether the murine antisera recognized Neisseria Heparin Binding Adhesin
(NHBA), which is one of the recombinant proteins in 4CMenB. Semchenko *et al.* [30] showed
that sera from 4CMenB-vaccinated individuals contains antibodies that cross-react with *Ng*
NHBA. Inspection of the western blots incubated with serum from 4CMenB-immunized mice
showed a band of approximately 66 kD, which correlates with the observed molecular weight
of NHBA. To determine whether this band was NHBA, we constructed an *nhbp* deletion
mutant and performed western blots against OMVs from this mutant and its isogenic parental
*Ng* strain, FA1090, using serum from 4CMenB-immunized mice. The results showed the
absence of the most intense band in this region of the blot in the mutant OMVs relative to
OMVs prepared from the parent strain (Fig 9). We conclude that the OMV portion of the
4CMenB vaccine induces cross-reactive antibodies against *Ng* PilQ, NHBA, and perhaps sev-
eral unidentified proteins based on similarities in apparent molecular weight in both humans
and mice, but that there is a species difference in the recognition of MtrE.

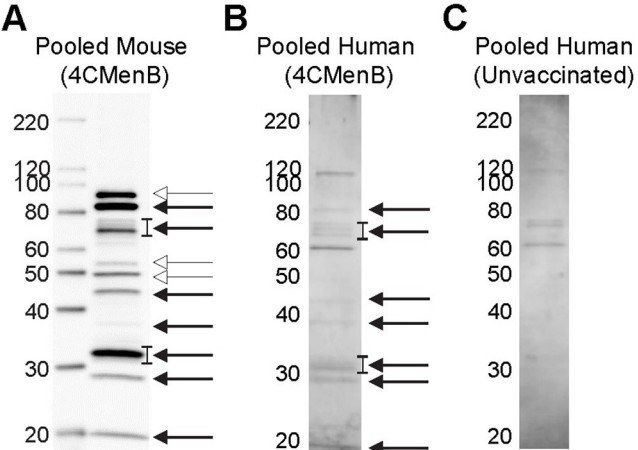

**Fig 7. Comparison of pooled serum from 4CMenB-immunized mice and humans for cross-reactivity against *Ng* OMV proteins.** The reactivity of pooled serum from 4CMenB-immunized mice and humans was tested for recognition of *Ng* strain F62 OMV proteins by western blot. Pooled serum from unvaccinated humans was tested in parallel. Mouse antiserum was collected 10 days after the third 4CMenB immunization; human subjects were immunized with 4CMenB two times and the sera were collected between 2 and 15 months after the second immunization as described in the Methods. **(A)** Pooled mouse serum from 4CMenB-immunized mice, 1:10,000 dilution; **(B)** pooled serum from 4CMenB-immunized persons, 1:500; **(C)** pooled serum from unimmunized persons, 1:500. Bands of similar molecular weight that were detected by both immunized murine and human sera are indicated by the solid arrow; bands that were detected only by mouse sera are indicated by the open arrow heads. The lowest band indicated by the open arrowheads is MtrE. The faint band under the 40 kD marker indicated by the solid arrow is consistent with the apparent molecular weight of PorB. The ~ 20 kD band that is recognized by both human and mouse serum is similar in apparent molecular weight to a band that was strongly recognized by serum from a 4CMenB-immunized individual in Semchenko *et al.* [30]. A band of this molecular weight was also intensely recognized by IgG and IgA in vaginal washes from 4CMenB-immunized mice (Fig 4B).

Bactericidal antibodies against one or more of these proteins may accelerate clearance of *Ng* in the mouse model as suggested by the presence of bactericidal activity in serum from immunized mice (Fig 3). Others have detected no or very low bactericidal activity in individuals vaccinated with 4CMenB- or NOMV-FHbp, which is an improved OMV-based serogroup B vaccine [30,40,41], while serum bactericidal activity against *Ng*, in contrast, was readily detected in immunized mice [40,41]. The *Ng* strain used in two of these studies was a highly serum-resistant strain, and therefore here we assessed the bactericidal activity of human serum from 4CMenB-immunized individuals against the serum-sensitive challenge strain used in our *in vivo* protection experiments. Seven of ten immunized individuals had a bactericidal$_{50}$ titer $\geq$ 1:960 against *Ng* strain F62 compared to and four of eleven unvaccinated individuals, but the difference was not statistically significant ($p \leq 0.3$, unpaired t test) (Fig 10). We conclude from these results that the contribution of bactericidal activity in 4CMenB-induced protection against *Ng* infection may differ between mice and humans; however, we did not directly investigate the role of complement in the *in vivo* protection we observed.

## Discussion

The pathogenic *Neisseria* are human-specific pathogens that differ in the capacity to cause life-threatening septicemia and meningitis (*Nm*) and nonulcerative sexually transmitted infections of the urogenital tract that can ascend to cause damage to the upper reproductive tract (*Ng*). The only reservoir for these pathogens is human pharyngeal, genital and rectal mucosae where the bacteria reside extracellularly and within an intracellular niche [42,43]. *Nm*, unlike *Ng*, produces a complex polysaccharide capsule that is critical for invasive disease, and vaccines that

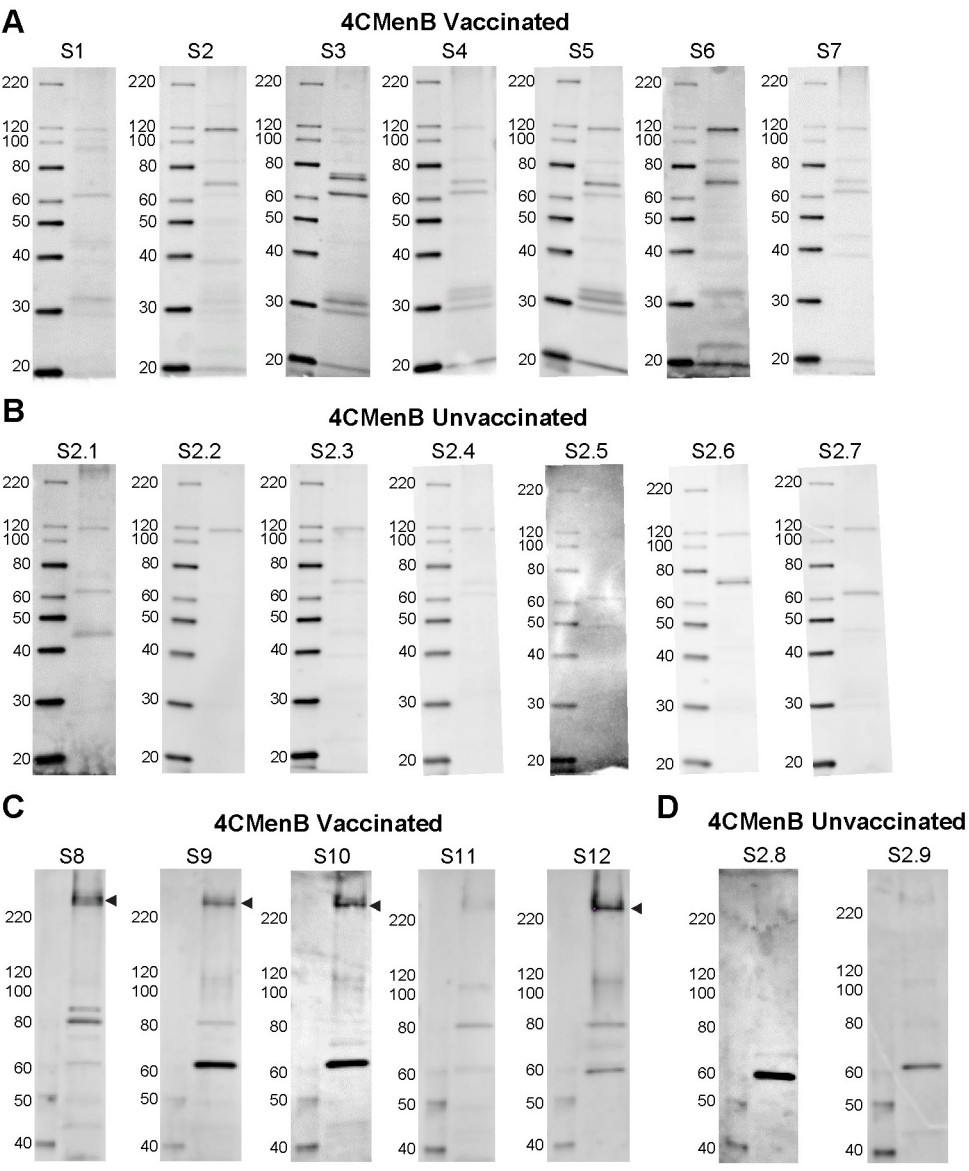

**Fig 8. Individual sera from 4CMenB-immunized subjects recognize several *Ng* OMV proteins including PilQ.**
Individual sera from **(A)** seven 4CMenB-immunized and **(B)** seven unimmunized individuals were tested against *Ng* F62 OMV by western blot. Several bands were recognized by sera from immunized people that were not detected by sera from unimmunized subjects with the exception of bands ~ 62 kD and 119 kD, which were detected by most sera regardless of immunization status. The reactivity of individual sera from **(C)** five 4CMenB-immunized and **(D)** two unimmunized individuals was tested against *Ng* strain F62 OMV proteins that were fractionated by SDS/PAGE for a longer period of time to allow visualization of PilQ. An intense band above the 220 kD marker, which is consistent with the migration of PilQ, was seen in blots incubated with four of the five immunized serum samples but not when incubated with two unimmunized control serum samples. Serum from the fifth immunized individual (S11) also recognized PilQ, but much less intensely.

target the capsules of four of the five most prevalent capsular serogroups A, C, W135 and Y have been effectively used for decades. This approach is not successful against serogroup B *Nm* due to the α2-8-linked polysialic acid composition of the serogroup B capsule, which mimics α2-8-sialylated human glycoproteins. A recent advance in biomedical research was the development of licensed vaccines that prevent serogroup B *Nm* invasive disease, one consisting of

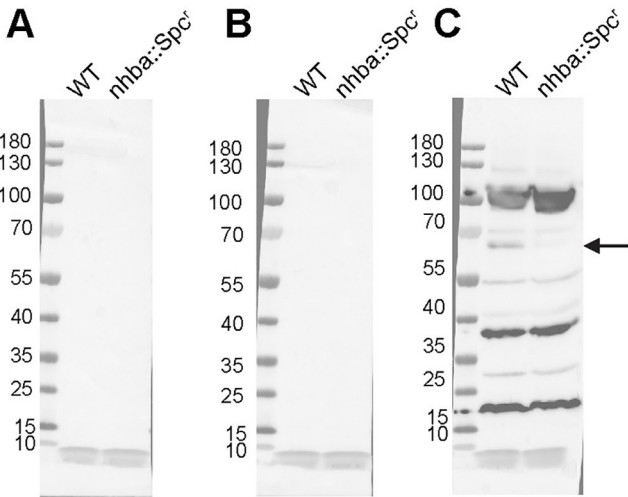

**Fig 9. 4CMenB-immunized mice have cross-reactive antibodies against *Ng* NHBA.** Crude OMVs (~10 μg) prepared from *Ng* strains FA1090 or FA1090-ΔNhba were fractionated by SDS-PAGE and transferred to nitrocellulose membranes. Western blot was performed using sera (1:2,500) from mice immunized with 4CMenB (right panel) or given alum alone (middle panel) or PBS (left panel). A band of ~ 66 kD is recognized by the immune serum in OMVs from the wild-type strain but not in OMVs from the isogenic *nhba* mutant FA1090-ΔNhba.

purified lipoprotein subunits, rLP2086 (Trumenba, Pfizer), and the other, 4CMenB (Bexsero, GSK) [44]. The 4CMenB vaccine was proceeded by *Nm* OMV vaccines that were tailor-made against endemic serogroup B strains in New Zealand, Brazil, Cuba and Norway [27,44].

Like vaccine development for serogroup B *Nm*, gonorrhea vaccine research has focused on conserved outer membrane proteins, *Ng* OMV, and in one case, the 2C7 oligosaccharide epitope within *Ng* LOS [18]. Vaccine development for gonorrhea is more complicated, however,

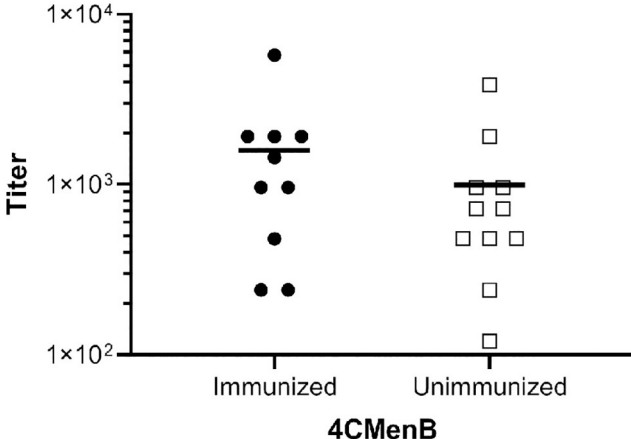

**Fig 10. Serum from 4CMenB-immunized and unimmunized individuals showed no significant difference in bactericidal activity against the challenge strain.** Human serum from ten 4CMenB-immunized and eleven unimmunized individuals was tested for bactericidal activity against *Ng* strain F62. The bactericidal$_{50}$ titer is shown for each individual (immunized: mean 1,584, median 1,200; unimmunized: mean 993, median 720). The difference in the two groups was not significant (unpaired t-test, p = 0.3). The interval between the second dose of 4CMenB and the date of serum collection ranged from 2 months (2 individuals), 4 months (6 individuals) and 15–16 months (2 individuals), with no potential correlation observed between number of months post-vaccination and bactericidal$_{50}$ titer.

due to a lack of defined immune correlates of protection. Meningococcal vaccine development is guided by serum bactericidal activity, which was identified as a correlate of protection for invasive infection in the 1960s, based on early detailed analyses of sera from case-control studies and different age groups [45]. Natural history studies for gonorrhea, in contrast, have identified only an association between antibodies against the restriction modifiable protein (Rmp) and increased susceptibility to infection, and in high-risk women, an association between antibodies to PorB and Opa proteins and reduced risk of *Ng* upper reproductive tract infection [18,19]. The lack of clear correlates of protection and the absence of immunity to reinfection by *Ng* have challenged the possibility of a gonorrhea vaccine.

It is in this context that the reported reduced risk of gonorrhea in individuals immunized with an *Nm* OMV vaccine [26] may herald a breakthrough for gonorrhea vaccine development. In support of this epidemiological evidence, we demonstrated that 4CMenB reproducibly accelerated *Ng* clearance and lowered the bioburden of *Ng* in a well-characterized mouse model of genital tract infection. 4CMenB induced vaginal and serum IgG1, IgG2a and IgA when administered subcutaneously that cross-react with several *Ng* OMV proteins expressed by six different *Ng* strains. These data are direct evidence of cross-species protection, and combined with our demonstration that several *Ng* proteins are recognized by sera from both 4CMenB-immunized mice and humans, support the use of female mice for studying vaccine-induced protection against gonorrhea.

As with many animal infection models differences in host factors including host-restricted factors must be considered when predicting whether the study outcome would occur in the natural host. Known host-restrictions that limit the capacity of mice to mimic human neisserial infections have been extensively reviewed [18] and include receptors for several neisserial colonization or invasion ligands and host scavenger proteins that are used by *Ng* to obtain iron [18] or zinc [46]. These restrictions limit the ability to fully evaluate the efficacy of vaccines in the mouse model that induce antibodies that block colonization or nutrient uptake although mice that are transgenic for human transferrin or carcinoembryonic antigen cellular adherence molecules (CEACAMs) could be used this purpose [47–49]. Host restrictions in two soluble negative regulators of the complement cascade, factor H (fH) and C4b-binding protein (C4BP), also exist and are especially important to consider when testing vaccines that clear *Ng* infection through bactericidal and opsonophagocytic activity. Transgenic hFH and hC4BP mice made for this purpose and not used in our study, were recently utilized by Rice and colleagues to more rigorously test the *in vivo* efficacy of immunotherapeutic strategies against *Ng* [50].

While unable to fully mimic human neisserial infections, animal models provide a physiologically relevant and immunologically intact system for testing vaccine-induced immune responses against infection. Meningococcal vaccine development has been aided by the use of mice and rabbits to test whether candidate vaccines cause adverse effects or induce bactericidal antibodies against *Nm* [51,52], and improved mouse and infant rat bacteremia models have been used to measure the efficacy of candidate *Nm* vaccines in eliminating *Nm* from the bloodstream [49,53,54]. More recently, infant rhesus macaques were employed to study immune responses to *Nm* OMVs, and are more predictive than mice in assessing whether these vaccines induce bactericidal antibodies against *Ng* in humans, perhaps due to species-specific differences in TLR-4 recognition of lipid A within these vaccines [40]. Early gonorrhea vaccine studies used chimpanzees, which do not have all the host restrictions found in other animal species, and human male subjects [18]. Chimpanzees are no longer used for gonorrhea research, however a human urethritis model is still available [55] and is the most relevant model for studying vaccine efficacy against *Ng* urethral infection in men.

Currently, the estradiol-treated mouse model is the only animal model for studying gonorrhea vaccine efficacy in females, where the majority of morbidity and mortality associated with gonorrhea occurs. This model is also used to systematically screen antigens, immunization regimens and adjuvants, and to analyze host responses [18]. Similarities between human and experimental murine infection include the fact that mice, like humans, produce a transient and unremarkable humoral response to *Ng* infection and can be reinfected with the same strain. *Ng* induces the Th17 pathway in both humans [19,20] and mice [56,57], which leads to recruitment of PMNs to the infection site. Similar to that reported for human cervical infections, hormonally driven, cyclical fluctuations in *Ng* colonization load and selection for Opa protein phase variants occurs over the course of murine infection [58]. *Ng* is seen within murine PMNs and importantly, *Ng* mutants that are more or less susceptible to killing by human PMNs or cationic antimicrobial peptides *in vitro* have a similar phenotype when tested against murine PMNs and cathelicidins, and are more fit or attenuated compared to the wild-type strain, respectively, during murine infection [58,59].

We also showed that the 4CMenB vaccine induced high titers of serum immunoglobulins that cross-reacted with *Ng* OMVs. Vaccine-induced vaginal IgA that recognized *Ng* proteins was readily detected by immunoblot. Recruitment of PMNs to the infection site occurred in all experimental groups, which could enable opsonophagocytic killing of *Ng* in the presence of specific antibody. Serum bactericidal activity was also detected that could mediate protection through complement-mediated bacteriolysis. Detailed investigation of the mechanism of protection have thus far only been reported for one candidate *Ng* OMV vaccine [60] and the 2C7 vaccine [35], both of which induced Th1 responses and bactericidal antibodies. The demonstration by Russell and colleagues that a vaginally applied Th1-inducing cytokine adjuvant clears *Ng* infection in mice and induces a specific adaptive response and protection from reinfection also suggests Th1 responses are protective [61,62]. Whether this is true for human infection is not known.

The importance of bactericidal activity in clearing *Ng* mucosal infections, is also not known. Passive protection studies with bactericidal monoclonal antibody against the 2C7 epitope clearly showed antibodies were sufficient for vaccine-mediated clearance in the mouse model [35]. However, this mechanism of protection may be vaccine specific. In our studies here, bactericidal activity against the serum-sensitive challenge strain was detected in pooled serum from immunized mice. A more definitive analysis, however, would be to determine whether the bactericidal activity of serum from individual mice correlates with clearance rate, which we were unable to do due to insufficient sample volumes. We did not detect a significant difference in the bactericidal$_{50}$ titers of serum from immunized humans compared to unimmunized individuals against the serum-sensitive challenge strain. If 4CMenB does indeed generate protective immune responses against *Ng* in humans, as is widely suspected based on ecological study data, these findings suggest that it is not due to bactericidal activity. A systematic assessment of the bactericidal activity of human immune serum against a diverse panel of *Ng* strains may be warranted, however, due to differences in susceptibility to complement and the expression level of surface components that could cause strain-specific variation in susceptibility to 4CMenB immune sera from humans. Further investigation of the immune responses induced by 4CMenB in mice and vaccinated humans will continue to define mechanisms of vaccine-mediated protection in both species.

Recently, the proteome of the 4CMenB vaccine was defined and shown to contain 461 proteins, of which 60 proteins were predicted to be inner membrane or periplasmic and 36 were predicted to be in the outer membrane or extracellular [63]. Others identified twenty-two *Nm* proteins as comprising >90% of the 4CMenB proteome, twenty of which have homologs in *Ng*, These investigators also showed that post-vaccinated serum from 4CMenB-immunized

humans recognized several bands in fractionated *Ng* OMVs by immunoblot [30]. In our study, the 4CMenB-induced antisera recognized several denatured OMV proteins in a panel of diverse *Ng* strains, which is consistent with the OMV portion of the 4CMenB vaccine generating a robust cross-reactive response. Recognition of LOS species was less impressive, with only ~25% of strains expressing LOS species that were reactive with anti-4CMenB serum. Using mass spectrometry and genetic deletion mutants, we identified several cross-reactive *Ng* proteins, including MtrE, PilQ, BamA, PorB, and NHBA, the gonococcal homolog of one of the recombinant components of 4CMenB. All of these proteins show promise as gonococcal vaccine targets [18,64].

The 94% amino acid identity (S1 Table) between the *Nm* and *Ng* homologs of MtrE identified in this report is consistent with the cross-reactivity that we observed. Importantly, the residues encompassing the two short surface-exposed loops of the MtrE monomer (residues 92–99 and 299–311 [65]), which are highly conserved among *Ng* strains, are identical to those from MtrE expressed by *Nm* strain MC58. MtrE is the outer membrane channel of the MtrCDE, FarABMtrE and MacABMtrE efflux pumps, which expel antibiotics and host-derived antimicrobial compounds [66]. The importance of the MtrCDE active efflux pump in protecting *Ng* against host innate effectors has been demonstrated in the mouse model [67]. Antisera directed against the two surface-exposed MtrE loops could target *Ng* for complement-mediated bacteriolysis and opsonophagocytosis, and may possibly impair efflux pump function to increase *Ng* susceptibility to host innate effectors. While serum from 4CMenB-immunized humans did not recognize MtrE by western blot, an MtrE subunit vaccine may induce a protective antibody response against MtrE in humans that is not generated through natural infection or when MtrE is provided to immune cells in the context of OMV, which contain many other antigens. The PilQ protein is critical for pilus secretion [68] and mutations in PilQ are associated with increased entry of heme and antimicrobial compounds [69] and enhanced resistance to cephalosporin [70]. Amino acids 406 to 770 of *Nm* PilQ were shown to be a promising vaccine target [71], and are 94% identical with the same region of *Ng* PilQ (strain FA1090). BamA is a surface-exposed outer membrane belonging to the Omp85 family [72,73]. The essential role of BamA in outer membrane protein biogenesis suggests it may be a highly effective vaccine target as it is present in cell envelopes and OMVs, surface-exposed, and well-conserved among clinical *Ng* isolates [38]. NHBA is an outer membrane protein that binds to heparin and other glycans and increases microcolony formation and adherence to human cervical and urethral cells [74]. *Ng* NHBA is well-conserved among *Ng* strains and 67% identical to the *Nm* NHBA present in the 4CMenB vaccine [64]. Although NHBA makes up a substantial portion of the immunizing antigen in 4CMenB, antibodies directed against the protein made up only a minor portion of the *Ng*-directed antibody responses in 4CMenB immunized mice. Immunization of mice with NHBA as a single antigen has been shown to induce bactericidal antibody responses as well as increased opsonophagocytic activity [64]. However, the relative contribution of anti-NHBA immunoglobulins induced by 4CMenB in protection against *Ng* still needs to be explored in mice and potentially humans.

In summary, the demonstration that a licensed *Nm* OMV-based vaccine accelerates *Ng* clearance in a murine genital tract infection model is direct evidence that cross neisserial species protection may be an effective vaccine strategy for gonorrhea. Whether this approach would protect against *Ng* rectal or pharyngeal infections, which are very common, is not known and in the absence of animal or human challenge models for these infections, this question must be solely addressed by epidemiological studies or clinical trials. Future detailed immunological studies in mice, which can be experimentally manipulated to directly test hypothesized mechanisms of protection, combined with clinical research studies on 4CMenB-

vaccinated humans should reveal new and important information on how to combat this ancient and highly successful pathogen.

## Methods

### Ethics statement

All animal experiments were conducted at the Uniformed Services University according to guidelines established by the Association for the Assessment and Accreditation of Laboratory Animal Care using a protocol approved by the University's Institutional Animal Care and Use Committee. Use of the human serum was approved by the USUHS Human Subjects Research Program Office. Serum specimens from the Department of Defense Serum Repository: The Armed Forces Health Surveillance Branch, Defense Health Agency, Silver Spring, Maryland [serum specimens 2016–2019; release dated 8/14/2019]. Human serum was collected as part of routine public health surveillance and therefore consent for this study was not available. However, samples provided for this work were deidentified (anonymous).

### Bacterial strains and culture conditions

*Ng* strains used in this study are listed in Table 2. To delete *nhba*, A DNA cassette for disrupting the *nhba* gene in *Ng* strain FA1090 was generated by synthetic gene synthesis. The disruption cassette contained ~400 bp upstream and downstream of FA1090 gene NGO_1958: bp 1934786–1935186 and bp 1936476–1936874 from the complete FA1090 genome sequence, and with the *Escherichia coli*-derived *aadA5* gene encoding spectinomycin resistance (Spc^r) [75], replacing the entire coding sequence of NGO_1958. *Ng* strain FA1090-ΔNhba was generated by spot transformation of the disruption cassette into of strain FA1090 as described by Dillard [76] followed by selection on GC agar plates containing spectinomycin (100 mg/L). Genomic DNA was prepared from Spc^r transformants, and the presence of the disrupted *nhba* locus was confirmed by PCR using primers spanning the coding region of *nhba* and within the Spc^r gene (S5 Fig). The presence of the Spc^r gene within the *nhba* locus was confirmed by nucleotide sequencing. Supplemented GC agar (Difco) was used to routinely propagate *Ng* as described [77]. GC-VNCTS agar [GC agar with vancomycin, colistin, nystatin, trimethoprim (VCNTS supplement; Difco) and 100 μg/ml streptomycin (Sm)] and heart infusion agar (HIA) were used to isolate *Ng* and facultatively anaerobic commensal flora, respectively, from murine vaginal swabs [78].

### Human serum

Serum from healthy human subjects was collected as part of ongoing Department of Defense public health serosurveillance [90]. Serum was requested from deidentified subjects who were previously immunized with the recommended 2 doses of 4CMenB, and were collected as part of an ongoing U.S. Defense Health Agency (DHA), Immunizations Healthcare Division (IHD) *in vitro* investigation of the human immune response against *Ng* after 4CMenB immunization (IHBISP_18_017). Serum specimens were collected between 2–16 months after the last 4CMenB immunization for the immunized samples. Sera from non-immunized controls were also available from the IHB-funded study.

### Immunizations and challenge experiments

Four-week-old female BALB/c mice (Charles River; NCI Frederick strain of inbred BALB/cAnNCr mice, strain code 555) were used in these studies. In pilot immunization studies, groups of 5 mice each were immunized with 20, 100 or 250 μL of 4CMenB (GSK) by the

**Table 2. Bacterial strains used in this study.**

| Strain | Source | Location, Date | Reference(s) |
|---|---|---|---|
| *N. meningitidis* | | | |
| MC58 | blood | UK, 1983 | ATCC [79] |
| *N. gonorrhoeae* (laboratory strains) | | | |
| F62 | urogenital | Atlanta, GA, 1962 | [80] |
| FA1090 | cervical (DGI) | Chapel Hill, NC, 1983 | [81] |
| **FA1090-ΔNhba** | - | - | This report |
| FA6140*pilQ*2 | *pilQ* mutant of FA1090 | | [82] |
| MS11 | cervical | Mt. Sinai, NY, 1972 | [83] |
| DW3-MS11 | *mtrE* mutant of MS11 | | [84] |
| FA19 | DGI | Copenhagen, 1959 | [85] |
| *N. gonorrhoeae* (clinical isolates) | | | |
| H041 | pharyngeal | Japan, 2009 | [86] |
| F89 | urethral | France, 2011 | [87] |
| LGB20[a] | urogenital | Baltimore, MD | [88] |
| LGB24[a] | urogenital | Baltimore, MD | [88] |
| WAMC 7720[b] | urethral | Fort Bragg, NC, 2014 | USU GC Isolate Reference Lab and Repository [89] |
| WAMC 7749[b] | urethral | Fort Bragg, NC, 2014 | As above |
| NMCP 4856[b] | urine | Portsmouth, VA, 2017 | As above |
| NMCP 9542[b] | urine | Portsmouth, VA, 2017 | As above |
| SAMMC 7363 | urine | San Antonio, TX, 2016 | As above |
| MAMC 3183 | urethral | Tacoma, WA, 2015 | As above |
| MAMC 3668 | urine | Tacoma, WA, 2019 | As above |
| NMCSD 3277 | urine | San Diego, CA, 2016 | As above |
| NMCSD 6364 | urethral | San Diego, CA, 2014 | As above |

PorB phenotype determined by Phadebact Monoclonal GC Assay (MKL Diagnostics). All strains are PorB1b except FA19, which is PorB1a.

[a]LGB20 and LGB24 were collected between 1991 and 1994 and were provided by Margaret Bash, CBER/FDA.

[b]WAMC isolates 7720 and 7749 were isolated 6 months apart in 2014 and NMCP isolates 4856 and 9542 were isolated 3 months apart in 2017.

intraperitoneal (IP) or subcutaneous (SQ) routes on days 0 and 28. Two independent immunization and challenge experiments were conducted. For these experiments, 250 μL of the vaccine were given IP or SC on days 0, 21 and 42. Control mice received PBS or alum in the form of Alhydrogel (InVivogen) diluted in PBS (n = 20–25 mice/group). Venous blood was collected on days 31 and 52; vaginal washes were collected on day 31. Three weeks after the final immunization, mice in the anestrus or the diestrus stage of the reproductive cycle were implanted subcutaneously with a 21-day slow-release 17β-estradiol pellet (Innovative Research of America) and treated with antibiotics to suppress overgrowth of potentially inhibitory flora as described [58]. Two days after pellet implantation, mice were inoculated vaginally with $10^6$ colony-forming units (CFU) of *Ng* strain F62. Vaginal swabs were quantitatively cultured for *Ng* on 7 consecutive days post-challenge and used to prepare stained smears to examine the influx of vaginal polymorphonuclear leukocytes (PMNs) [78].

## Enzyme-linked immunosorbent assay (ELISA) and western blots

Serum or vaginal total Ig, IgG1, IgG2a and IgA were measured as endpoint titers as determined by standard ELISA [91]. Microtiter plates were coated with 20 μL/well of a 1:5 dilution of the formulated Bexsero vaccine in 15 mM $Na_2CO_3$, 35 mM $NaHCO_3$, pH 9.5, or with 4 μg/

ml of OMV from *Ng* strain F62. OMVs were isolated from supernatants from late-logarithmic phase cultures that were centrifuged for 1 hour at 100,000 x g at 4˚C. Pellets were resuspended in 1 mL of PBS. Protein concentration was determined by the BCA protein assay (Thermo Scientific). For whole-cell lysates (WCL) (total cellular proteins), bacteria from agar plates or mid-logarithmic phase cultures were centrifuged and the bacterial pellets suspended to an $OD_{600}$ = 0.5. One milliliter of this suspension was mixed with 60 μL of Laemmli sample buffer. For western blots, WCL (4 μL) or 20 μg of OMV were subjected to sodium dodecyl sulfate polyacrylamide gel electrophoresis (SDS-PAGE), transferred to nitrocellulose membranes, stained with Ponceau S, and blocked overnight with 0.5% Tween20 in PBS. Membranes were incubated with pooled antisera or vaginal washes from each experimental group diluted in block and washed three times with 0.05% Tween 20 in PBS. Secondary antibody was horseradish peroxidase (HRP)-conjugated anti-mouse IgG or IgA and a chemiluminescence HRP was used as substrate (GE Healthcare). For western blots with human serum, immunized and unimmunized sera were diluted 1:500; secondary antibody was horseradish peroxidase (HRP)-conjugated anti-human IgG. Apparent molecular weight of bands was determined using a standard curve generated from molecular weight markers ($r^2$ = 0.9916). For western blots with LOS, proteinase K-treated bacterial extracts were generated as described [92] without the phenol treatment step, separated on 16% tricine gels (Novex) and probed with 1:10,000 dilutions of pooled serum from immunized and control mice followed by HRP-conjugated anti-mouse IgG as above.

## Immunoprecipitation and mass spectrometry

Two milliliters of a *Ng* suspension ($OD_{600}$ = 1) prepared from a mid-logarithmic phase Gc broth culture were mixed with 30 μL of antisera for 20 minute at room temperature. Cell pellets were washed once with GCB and solubilized in 2% Zwittergent 3,14 (EMD Millipore) in PBS for one hour at 37˚C. The solubilized suspension was centrifuged for 10 minutes at 2,000 x g, the supernatant mixed with protein A/G resin (ExAlpha Biologicals) for two hours at 4˚C with mixing, and the resin washed three times with 0.5% Zwittergent 3,14 in PBS, and once with PBS alone. The resin was suspended in 50 μL Laemmli sample buffer without β-mercaptoethanol and subjected to SDS-PAGE for Western blotting; a duplicate gel was run in parallel and stained with Coomassie G-250. Bands from the stained gel were submitted to the Michael Hooker Proteomics Center at the University of North Carolina at Chapel Hill for trypsin digest and identification using mass spectrometry. Accession ID numbers of proteins described in this report are disclosed in Table Table 1 and S1 Table. Alignment of amino acid sequences was performed using ClustalW.

## Bactericidal assay

A modification of a previously described bactericidal assay [93] was used to test the bactericidal activity of mouse and human serum. Pooled sera from immunized and control mice were heated at 56˚C for 30 min and serially diluted 1:2 in minimal essential medium (MEM) (1:30–1:960). Fifty microliters of each dilution were pipetted into wells of a 96-well microtiter plate. Fifty microliters of an MEM suspension containing 100–400 CFU of the target strain were added to the wells and to a well containing 50 μl of MEM alone. After 5 minutes incubation at RT, 50 μl of pooled normal human serum (NHS) (PelFreeze) were added to each well (final concentration 10%) and the plate was incubated for 55 minutes at 37˚C in 5% $CO_2$. Fifty microliters of GC broth were then added, mixed, and 50 μl aliquots were cultured in duplicate on GC agar and incubated overnight. The antiserum dilution that gave 50% recovery compared to wells without antiserum was defined as the bactericidal$_{50}$ titer. Wells containing heat-

inactivated NHS were tested in parallel to measure complement-independent loss of bacterial viability during the assay; no appreciable loss was detected in any experiment. Human sera were tested using the same assay, with pooled sera from five 4CMenB-immunized subjects and five unimmunized subjects tested initially, followed by bactericidal assays with individual sera from ten immunized and eleven unimmunized subjects. The assays were performed against each strain in two independent experiments; a third assay was performed for those titers that differed in the first two experiments, and the titer that was identified in two of the three iterations was used.

## Statistical analysis

ELISA titers were compared by a Kruskal-Wallis test with Dunn's multiple comparison. For challenge experiments, the percentage of mice with positive cultures at each time point was plotted for each experimental group as a Kaplan Meier curve and analyzed by the Log Rank test. The number of CFU recovered from vaginal swabs over time was compared by repeated measures ANOVA with Bonferroni correction. The area under the curve (AUC) was calculated for each individual mouse by determining the AUC across the 7 culture time points that was above the limit of detection (20 CFU/mL). Differences between AUC and percentage of vaginal PMNs were compared using a Kruskal-Wallis test with Dunn's multiple comparison. Differences in bactericidal$_{50}$ titers were assessed by unpaired t and Mann Whitney tests. Statistical analyses were performed using the software Prism (GraphPad Software, La Jolla, CA). Raw data used for statistical analysis of ELISA, *in vivo* efficacy testing, and mouse bactericidal assays have been published [94].

## Supporting information

**S1 Table. Amino acid identity between proteins of *N. meningitidis* MC58 and *N.** (DOCX)

**S1 Fig. Pilot dose response immunization study with the 4CMenB vaccine.** Groups of 5 BALB/c mice were given 20, 100 or 250 μl of the formulated vaccine on days 1 and 28 by the IP or SC routes. **(A,B)** Serum IgG1 and IgG2a titers using microtiter plates coated with the formulated 4CMenB vaccine 10 days after the second immunization. A dose response is shown for serum IgG1 in both IP- and SC-immunized mice. **(C)** Serum reactivity (1:5,000) against whole cell lysates of *Nm* strain MC58 and of 6 different *Ng* strains using anti-mouse IgG secondary antibody shows a similar dose response based on differences band intensity. A nonparametric test (Kruskal Wallis with Dunn's multiple comparison) was used to analyze ELISA data due to the low sample size. **, p < 0.01. (TIF)

**S2 Fig. 4CMenB significantly accelerated *Ng* clearance and reduced the *Ng* colonization load in two independent experiments.** In each experiment, mice were immunized three weeks apart with 250-μl doses of 4CMenB by the IP (blue) or SC (red) route or given PBS (black) or alum (purple) by the IP route (n = 25 or 20 mice per group in experiments 1 **(A-C)** and 2 **(D-F)**, respectively). Three weeks after the final immunization, mice were challenged with *Ng* strain F62 as described in the Methods. **(A,D)** Percentage of culture-positive mice over time and average CFU per ml of a single vaginal swab suspension, respectively for experiment 1 (n = 20–23 mice/group); **(B,E)** Percentage of culture-positive mice over time and average CFU per ml of a single vaginal swab suspension, respectively for experiment 2 (n = 18–19 mice/group); **(C,F)** Total bioburden recovered from individual mice over 7 days expressed as

area under the curve (AUC) for experiments 1 and 2, respectively. The median with 95% CI is indicated for each group.*, p < 0.05, **p < 0.01, *** p < 0.0001.
(TIF)

**S3 Fig. A vaginal PMN influx occurred in all experimental groups.** Vaginal smears collected on each culture day following bacterial challenge were stained with Hemacolor Stain (Sigma), and the percent of PMNs among 100 vaginal cells was determined by cytological differentiation using light microscopy. An increase in the percentage of PMNs occurred between days 4–7 as is characteristic of this model, with no statistical difference between the groups. The median percent PMNs is shown by the horizontal bar for each time-point.
(TIF)

**S4 Fig. 4CMenB antiserum recognizes *Ng* LOS in a minority of strains.** Proteinase K-treated bacterial extracts from 4 laboratory strains and 13 clinical isolates were resolved on 16% Tricine gels and stained with silver stain (top blot, panel A) or Emerald green (top blots, panels B and D), or electroblotted and probed with the following: **(A)** Mabs 3F11, 4C4 and 2C7, which recognize *Ng* LOS epitopes (bottom panels). Note that FA19 1986 is a variant of FA19 (panel B) that has a phase-off *lgtA* gene that results in truncation of the LOS to a single 3.6 kDa species [95]. **(B)** Pooled IP250 4CMenB antisera (bottom panels). The doublets in LGB-24 and NMCSD 3277 and single LOS species in MS11 and NMCSD 6364 that were recognized by the antiserum are distinct from the LOS species identified by the Mabs shown in Panel A. **(C)** Western blots against LOS from strains with 4CMenB-reactive bands incubated with pooled sera from 4CMenB-immunized mice versus mice given Alum only. **(D)** Emerald green-stained (upper panel) LOS from *Ng* strain H041 and LGB-24 (positive control) used to inoculate mice (Inoc) and from vaginal cultures collected on days 2 and 5 of infection. No change in the LOS species or 4CMenB reactivity was observed during infection by these strains.
(TIF)

**S5 Fig. Disruption of *nhba* in *Ng* strain FA1090. (A)** Schematic of the synthetic *Ng nhba* disruption cassette showing 400 bp homology arms 5' and 3' from the *nhba* Ngo_1958) coding sequence, which is completely removed and replaced with a Spc$^r$ gene. The approximate location of primers used for polymerase chain reaction to detect the disrupted gene and he predicted size of the amplification products is noted. **(B)** Disruption cassette (Lane 1), and genomic DNA from FA1090 (Lane 2) and FA1090-Δ*nhba* (Lane 3) were used as template for PCR with the indicated primers. The products of the amplification reactions were run on 1X TBE agarose gels and visualized with ethidium bromide staining. Oligonucleotide sequence primers were: Primer 1: ACGTTTTGTTTACCGCTGCC; Primer 2: TTCGGGGGCTTGTT TGATGA; Primer 3: CGTTGTCCCGCATTTGTAC.
(TIF)

## Acknowledgments

The authors wish to thank Thomas Hiltke, Carolyn Deal and Leah Vincent for helpful discussions, Ian Feavers and Carolyn Vipond for technical advice on ELISAs using 4CMenB as the coating antigen, Melissa Samo for conducting the ELISAs early in this study, and James E. Anderson for providing *Ng* mutant strains, protocols and helpful suggestions. We are also grateful to Rachel Rowland for preparation of media and assistance with animal handling. Mass spectrometry services were provided by the Michael Hooker Proteomics core facility at the University of North Carolina at Chapel Hill.

## Disclaimer

The content of this article is solely the responsibility of the authors and does not necessarily represent the official views of the Department of Veterans Affairs, the Department of Defense, the Uniformed Services University or the NIH.

## Author Contributions

**Conceptualization:** Isabelle Leduc, Ann E. Jerse.

**Data curation:** Kristie L. Connolly.

**Formal analysis:** Isabelle Leduc, Ann E. Jerse.

**Investigation:** Isabelle Leduc, Kristie L. Connolly, Afrin Begum, Knashka Underwood, Stephen Darnell, Jacqueline T. Balthazar, Andrew N. Macintyre, Joseph A. Duncan, Marguerite B. Little, Nazia Rahman, Eric C. Garges.

**Methodology:** Isabelle Leduc, William M. Shafer, Gregory D. Sempowski, Ann E. Jerse.

**Supervision:** Kristie L. Connolly, Ann E. Jerse.

**Writing – original draft:** Isabelle Leduc, Kristie L. Connolly, Ann E. Jerse.

**Writing – review & editing:** William M. Shafer, Gregory D. Sempowski, Joseph A. Duncan, Eric C. Garges, Ann E. Jerse.

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
