## [Decision Letter · Decision Letter 0]

7 Jun 2020

Dear Ann,

Thank you very much for submitting your manuscript "The serogroup B meningococcal outer membrane vesicle-based vaccine 4CMenB induces cross-species protection against Neisseria gonorrhoeae" for consideration at PLOS Pathogens. As with all papers reviewed by the journal, your manuscript was reviewed by members of the editorial board and by several independent reviewers. The reviewers appreciated the attention to an important topic. Based on the reviews, we are likely to accept this manuscript for publication, providing that you modify the manuscript according to the review recommendations.

Reviewer 2 raises important issues relating to how the results described here relate to immune responses detected in humans following immunisation with Bexsero (reported by Semchenko et al.,). Further comments about the differences and similarities are warranted.

best,

Chris

Christoph Tang

Section Editor

PLOS Pathogens

Christoph Tang

Section Editor

PLOS Pathogens

Kasturi Haldar

Editor-in-Chief

PLOS Pathogens

orcid.org/0000-0001-5065-158X

Michael Malim

Editor-in-Chief

PLOS Pathogens

orcid.org/0000-0002-7699-2064

Reviewer 2 raises important issues relating to how the results described here relate to immune responses detected in humans following immunisation with Bexsero (reported by Semchenko et al.,). Further comments about the differences and similarities are warranted.

Reviewer Comments (if any, and for reference):

Reviewer's Responses to Questions

**Part I - Summary**

Reviewer #1: Leduc et al show that immunizing mice with MenB-4C (Bexsero) results in attenuation of gonococcal vaginal colonization. Pooled post-dose 3 immune serum showed bactericidal activity against strains F62 and FA1090. Ng proteins (eg, MtrE, BamA and PilQ) recognized by IgG from immune mouse serum have been identified. This is an important study in light of the 31% clinical efficacy of the MeNZB vaccine against gonorrhea in New Zealand. How MeNZB (and possibly the related vaccine, Bexsero) may act to provide some immunity against gonorrhea remains unclear. This paper uses a mouse model of gonorrhea to address this issue. The presentation of some of the results needs further clarification. The discussion also needs to be modified to take into consideration related work in the field and tie together observations and also highlight the differences between this study and recent studies of Bexsero immunogenicity versus N. gonorrhoeae in humans and macaques.

Reviewer #2: This manuscript describes the results of vaccine studies using Bexsero in an estradiol treated mouse model of vaginal gonorrhea. The manuscript is well written and presents the data in a detailed fashion. It is clear that considerable effort and thought has gone into the design of the experiments. The most important question which needs to be addressed is how closely this model simulates human disease and can be murine response be utilized to allow selection of antigens for a successful human gonococcal vaccine. This reviewer believes that the authors of this manuscript have failed to justify the results the obtained in this regard.

Reviewer #3: It is seldom that I have a chance to review a paper that is a real pleasure to read. This manuscript describes an extremely important, in some ways game-changing, study related to the development of vaccines against Ng (N. gonorrhoeae), a goal that has been elusive despite decades of study. The genesis of this paper is the observation of efficacy against Ng infection in a field trial of a vaccine against Nm (N. meningitides). These authors have taken that observation into the laboratory by demonstrating similar protection in a murine mouse model of N. gonorrhoeae infection developed by the senior author some time ago. This one-two punch is a major cause for optimism in the quest for a vaccine, one because, feasibility of a vaccine approach in a clinical trial has been demonstrated, and two because this model provides the opportunity to dissect mechanisms of protection, and to evaluate new vaccine candidates prior to human trials. The paper is very well written, and the experiments proceed logically. In fact, as I read, I found that experiments I was wanting to see, were in fact included already. To restate the significant findings of this paper: 1) A human Nm vaccine found to provide some efficacy against Ng in a clinical trial, was found to do the same in a mouse model of Ng infection. 2) Outer membrane protein candidate antigens identified by mouse immune sera illuminate paths forward for Ng vaccine development. 3) antibodies against LOS do not appear to be involved in mediating protection. 4) The mouse model has been validated as a tool for study of immunity to Ng infection. An additional aspect of this paper was the quality of the introduction and the discussion. Both nicely put this work in historical perspective in a concise and clear manner.

**Part II – Major Issues: Key Experiments Required for Acceptance**

Reviewer #1: (No Response)

Reviewer #2: A.The mice in the current study showed significant bacteridical activity against both serum sensitive and serum resistant gonococcal strains (Figure 3). In the Semchenko study, humans immunized with Bexsero did not develop a bactericidal response to the gonococcus. While it can be difficult to compare, the intensity of the immune response both in the number of antigens and types recognized in the murine response was considerably greater both by western blot and ELISA measurements than the human Bexsero response. The gonococcal literature since the mid-1970s has been replete with studies that have shown murine immune responses to pilus, porin and LOS epitopes which the human immune system either cannot or does not recognize. It would appear comparing the results of the Semchenko paper to this manuscript that we may be seeing the same situation in this model.

B. For the results obtained in this model to be accepted as valid surrogates of the human response, it would be necessary to take the results from the murine system and demonstrated they have meaning in a human system. One possible way to do this would be to demonstrate that an anti-mtrE response is a significant part of the human response to Bexsero. Semchenko demonstrated a robust Neisseria Heparin binding antigen response in Bexsero immunized human sera by ELISA. The authors should consider taking at least one of the three antigens which appear to be potential vaccine candidates in the mouse model and show that antibodies to it, are present in Bexsero immunized human sera. Wang and Associates (J. Infection, 2018, 71:191-224) showed that mice immunized with mtrE surface exposed peptides showed protection in the same murine model of gonococcal infection. Using these peptides, the authors could address this issue directly if they want to prove that their model useful in identifying potential human vaccine candidates.

Reviewer #3: As stated in the summary above, I feel that all of the major points and conclusions of this paper are well supported by the experimental evidence. I did not identify any necessary additional experiments.

**Part III – Minor Issues: Editorial and Data Presentation Modifications**

Reviewer #1: 1. The graphs in Fig 3 should show the killing over the entire range of concentrations – 1:30 to 1:960 (note the graph shows a range of 120 to 1920, while methods mention dilutions from 1:30 to 1:960, so need to unify). Please also show each value at least for the samples that got NHS – the methods state 2-3 independent experiments. Were the differences statistically significant? Also, what do the data points represent – mean, median?

2. The reason for pooling sera for bactericidal assays should be stated. This limitation will overlook heterogeneity in bactericidal responses across animals and should be discussed.

3. The authors have made a considerable effort to look for antibody responses to LOS (Fig 5). The OMVs in Bexsero are detergent-extracted, therefore contain only trace amounts of LOS. The anti-LOS cidal Ab described in Ref 32 were elicited after immunization with LPS-containing vaccine (NOMVs or detoxified LPS conjugated to proteins). The anti-LPS cidal Ab seen in children who got the Chilean vaccine (detergent-extracted OMV, contains <1% LPS) were seen 6 or more months after the last dose of vaccine, suggesting that they developed as a result of subsequent meningococcal colonization. Thus, rationale provided for looking at LOS response (lines 217-8) is unclear. It is not surprising that there is a limited response to LOS (from the presented data its not discernable whether pre-immune or adjuvant sera also would have reacted with Ng LOS). Note that humans immunized with Bexsero barely mount an Ab response to meningococcal or Ng LOS over baseline (see Supp Fig 6 in Semchenko et al, Clin Infect Dis, Vol 69, Issue 7, p 1101–1111). I suggest this entire section on LOS be removed or stated as a single sentence. The corresponding sentence in the abstract (lines 46-7) should also be deleted and Table 2 modified accordingly.

4. The discussion focuses excessively on justification of use of this mouse model, and the possible advantages of transgenic mice. These portions (lines 332-372) can be considerably shortened and some of the comprehensive, outstanding reviews by the authors cited instead. The discussion should instead focus on how these results relate to other similar work (please see below).

5. There should be a discussion of how these results compare to results reported by Semchenko et al, who examined Ng cross-reactive antibodies in humans given Bexsero. Were similar Ng proteins identified by immune mouse and human sera? Agreeably, the cross-reactive proteins, with the exception of NHBA, were not fully characterized in that study but it would be worth commenting on differences in molecular weights of the target proteins. It appears that the major Ng protein recognized in Bexsero immunized humans is in the ~25 kD range and against NHBA but does not appear to be the case here. These mouse/human differences are important and merit discussion.

6. Minimal bactericidal activity versus FA1090 was observed with serum from macaques immunized with Bexsero when tested at a 1:5 dilution (Beernink et al, mBio. 2019 Jun 18;10(3):e01231-19.). Humans given MenB-4C also do not show bactericidal activity against FA1090 (Beernink et al. J Infect Dis. 2019 Apr 1; 219(7): 1130–1137). Possible reasons for differences in the serum bactericidal responses across the species should be discussed.

7. Lines 386-389 “…; however, eleven other promising…. (..unpublished data…)”. This part of the sentence should either be substantiated with data or deleted. This reviewer acknowledges that vaccines may function through mechanisms other than complement-dependent killing. Reports of high cidal titers need to be interpreted cautiously. This is because mice and rabbits often develop natural bactericidal Abs vs Ng with age and bactericidal assays reported for gonococcal vaccine candidates sometimes lack critical age-matched adjuvant controls, and thus does not account for the effect of these natural cidal Ab (eg, PMID: 27141096; PMID: 27895130). This study demonstrates baseline bactericidal activity in the pooled serum from unimmunized age-matched mice – a key control.

8. Line 71 – 87 million new infections (Rowley et al, Bull World Health Organ 2019;97:548–562P)

9. Line 127 – NHBA (not NHBP)

10. Was the NHS from Pel-Freez IgG and IgM depleted?

11. What dilution of serum was used in S1 Fig, panel C?

12. Please also show the Area Under Curves for each individual mouse experiment (S2 Fig).

13. Minor details/discrepancies in cidal methods in methods vs Fig legends – methods state 100-400 CFU used, legend states 10^4 bacteria. Time of incubation – 55 min in methods, 45 min in Fig legend.

14. Line 316 –The meningococcal group C outbreak described in Ref 44 occurred at Fort Dix, NJ (not Fort Ord, CA).

15. Lines 328-329 – only the 3 doses were similar, not the regimen. The vaccine preparation and the dosing interval were both different. Suggest deleting this sentence.

Reviewer #2: (No Response)

Reviewer #3: I have no additional editorial suggestions.

PLOS authors have the option to publish the peer review history of their article (what does this mean?). If published, this will include your full peer review and any attached files.

Reviewer #1: No

Reviewer #2: Yes: Michael A. Apicella, M.D.

Reviewer #3: No
---

## [Editor Report · Decision Letter 1]

8 Oct 2020

Dear Dr Jerse,

We are pleased to inform you that your manuscript 'The serogroup B meningococcal outer membrane vesicle-based vaccine 4CMenB induces cross-species protection against Neisseria gonorrhoeae' has been provisionally accepted for publication in PLOS Pathogens.

Best regards,

Christoph Tang

Section Editor

PLOS Pathogens

Christoph Tang

Section Editor

PLOS Pathogens

Kasturi Haldar

Editor-in-Chief

PLOS Pathogens

orcid.org/0000-0001-5065-158X

Michael Malim

Editor-in-Chief

PLOS Pathogens

orcid.org/0000-0002-7699-2064
---

## [Editor Report · Acceptance letter]

28 Nov 2020

Dear Dr Jerse,

We are delighted to inform you that your manuscript, "The serogroup B meningococcal outer membrane vesicle-based vaccine 4CMenB induces cross-species protection against Neisseria gonorrhoeae," has been formally accepted for publication in PLOS Pathogens.

Best regards,

Kasturi Haldar

Editor-in-Chief

PLOS Pathogens

orcid.org/0000-0001-5065-158X

Michael Malim

Editor-in-Chief

PLOS Pathogens

orcid.org/0000-0002-7699-2064